# New Selective Progesterone Receptor Modulators and Their Impact on the RANK/RANKL Complex Activity

**DOI:** 10.3390/molecules25061321

**Published:** 2020-03-13

**Authors:** Katarzyna Błaszczak-Świątkiewicz

**Affiliations:** Department of Applied Pharmacy, Medical University of Lodz, Muszynskiego 1, 90-151 Lodz, Poland; katarzyna.blaszczak-swiatkiewicz@umed.lodz.pl; Tel.: +48-426779240

**Keywords:** RANK-RANKL complex, antiprogestins, mesoprogestins, osteoporosis, breast cancer

## Abstract

Breast cancer depends on women’s age. Its chemotherapy and hormone therapy lead to the loss of bone density and disruption of the skeleton. The proteins RANK and RANKL play a pivotal role in the formation of osteoclasts. It is also well established that the same proteins (RANK and RANKL) are the main molecules that play an important role in mammary stem cell biology. Mammary stem cells guarantee differentiation of the epithelial mammary cells, the growth of which is regulated by the progesterone-induced RANKL signaling pathway. The crosstalk between progesterone receptor, stimulated by progesterone and its analogues results in RANKL to RANK binding and activation of cell proliferation and subsequently unlimited expansion of the breast cancer cells. Therefore downstream regulation of this signaling pathway is desirable. To meet this need, a new class of selective estrogen receptor modulators (SPRMs) with anti- and mesoprogestin function were tested as potential anti-RANK agents. To establish the new feature of SPRMs, the impact of tested SPRMs on RANK-RANKL proteins interaction was tested. Furthermore, the cells proliferation upon RANKL stimulation, as well as NFkB and cyclin D1 expression, induced by tested SPRMs were analyzed. Conducted experiments proved NFkB expression inhibition as well as cyclin D1 expression limitation under asoprisnil and ulipristal treatment. The established paracrine anti-proliferative activity of antiprogestins together with competitive interaction with RANK make this class of compounds attractive for further study in order to deliver more evidence of their anti-RANK activity and potential application in the breast cancer therapy together with its accompanied osteoporosis.

## 1. Introduction

Breast cancer is a disorder which depends on the women’s age and hormonal system activity [1]. Estradiol determines the growth of mammary stem cells. It is well established that receptor activator of nuclear factor κB (RANK, the receptor for RANKL) and receptor activator of nuclear factor kappa-Β ligand (RANKL) are the main molecules that play an important role in mammary stem cell biology. Differentiation of epithelial mammary cells is regulated by the progesterone-induced RANKL signaling pathway. Crosstalk between progesterone receptor, stimulated by progesterone, is expressed by the interaction between the progesterone receptor and RANKL, results in RANKL to RANK binding and activates cell proliferation and subsequently unlimited expansion of the breast cancer cells. Moreover, the same RANK-RANKL complex plays a pivotal role in osteoclastogenesis [2]. Osteoporosis often accompanies breast cancer treatment with selective estrogen receptor modulators (SERMs) and is a common disease, especially among menopausal women [3]. Estradiol deficiency during menopause or surgical removal of the ovaries is responsible for the loss of bone density in the female body. Although a similar bone density reduction effect is observed in correlation to low testosterone levels in the male body, osteoporosis is more common for women than men. The statistical analysis proves about 15% of people in their 50 s and 70% of those over 80 have osteoporosis [4]. The expression of RANK-RANKL molecules goes through two biological response pathways: directly by interaction with RANK/RANKL and indirectly by expression of the progesterone receptor. The first way is based on the activity of osteoprotegerin (OPG) as a decoy receptor for RANKL. The second one is an interaction between the progesterone receptor (PR) ligand and progesterone. The osteoprotegerin expression is induced by female sex hormones and inhibits the development and activation of osteoclasts, which in the long run results in the inhibition of bone loss. OPG is the main regulator of the crosstalk between estrogen and RANKL expression referring to the bone destruction repression [2]. To prevent the overexpression of osteoclasts in osteoporosis, the new target—RANK—and RANK/RANKL complex as well as progesterone receptor and their biological function should be addressed as a new aspect of prevention of women’s health disorders during their postmenopausal and hormone-depend time of life. Some alternative ways of RANK-RANKL signaling have been discovered and described in the [5,6]. Transcriptional NF-κB factor normally exists in the cytoplasm as an inactive complex with nuclear factor of kappa light polypeptide gene enhancer in B-cells inhibitor, alpha (IκBα). Phosphorylation of IκBα by IKK complex (IκB kinase, IKK, is an enzyme complex) leads to complex degradation and free NFkB migrates to the nucleus and triggers DNA. Transcription of specific genes ensures i.a osteoclast formation and also protects cells against apoptosis like in breast cancer expansion [7,8,9,10,11,12,13]. Given the NF-κB function approach as well as its relation to osteoporosis and breast cancer, it is obvious why the recognition of the NF-κB activity in the crossway between RANK/RANKL and SPRMs is crucial to provide the evidence of useful SPRMs as an anti-RANKL and antiproliferative agents into osteoporosis and breast cancer treatment. The aim of planed studies was to initiate the experiments referring to the role of tested compounds into NFkB function, especially in breast cancer and osteoporosis progress. Ulipristal and Asoprisnil have been tested as potential ligands for the RANK receptor. In this study, the effect of SPRMs on RANKL-induced signaling has been examined. The structures of the tested compounds are shown in Table 1.

## 2. Results and Discussion

### 2.1. Proliferation Assay

Hormone replacement therapy (HRT) with combination therapy of estradiol and progesterone brings breast cancer risk for women [13]. Therefore to eliminate the negative aspect of this therapy, the role of progesterone—the main factor that triggers RANKL expression and breast cancer cells proliferation—needs to be limited. The first test that allowed us to estimate the influence of steroid ligands and OPG into RANKL induced proliferation of T47D cells was WST assay. The control cells proliferation under RANKL stimulation was detected and compared with negative control (RANKL-). Osteoprotegerin (OPG) is the main regulator of the crosstalk between estrogen and RANKL expression therefore OPG was used as a positive control. When the OPG level goes down, the RANKL activity increases and results in cells proliferation. The test system was treated by OPG and increased concentration of mifepristone and progesterone (the highest tested concentration is shown in Figure 1). The tested cell viability upon OPG stimulation proved their proliferation inhibition in the presence of RANKL. A similar effect was obtained using mifepristone. Although mifepristone showed antiproliferation activity within RANKL- cells, its anti-RANKL activity towards RANKL+ cells is significant and opposite to the strong proliferation of progesterone (Figure 1). A positive aspect of antiproliferation activity of mifepristone in RANKL-induced cells opened a new perspective for SPRMs function. Therefore subsequent tests were extended to new anti- and mesoprogestin in order to compare their activity.

### 2.2. NFkB Expression Assay

The receptor activator of NF-kB-ligand (RANKL) is involved in osteoclast differentiation in the course of osteoporosis and required for mammary cancer development. The engagement of progesterone in cell proliferation takes place through two pathways: an endocrine one via interaction with progesterone receptor in PR+ cells and a paracrine one via elicitation of a mitogenic effect in PR- cells with the activity of tumor necrosis factor (TNF) family member – receptor activator of NF-κB-ligand (RANKL). RANKL induces the proliferation of PR- cells and ensures the PR- cells phenotype survival. Therefore the search for new potential inhibitors of RANK/RANKL signaling of both genomic and non-genomic pathway is desired [14].

This genomic mechanism of action was tested on PR-positive T47D cells stably transfected with NFkB vector linked to LUC reporter vector. The luminescence signal was measured so as to assess NFkB expression upon SPRMs induction. Mifepristone appeared as a strong anti-RANK agent with 44-fold inhibition of NFkB expression. Also, ulipristal proved its inhibition towards NFkB expression. Its anti-RANK activity was weaker than mifepristone but almost seven times stronger than promegestone (Figure 2B). Both of these antiprogestins showed dose-depend inhibition of NFkB expression (Figure 2A). Mifepristone and ulipristal efficacy compared to promgestone at EC_70_ was equal to 5% and 27%, respectively. These outcomes proved the potential use of antiprogestins as anti-RANK agents. Also a mesoprogestin—asoprisnil—presents potential activity as an anti-RANK agent. its efficacy for EC_70_ although it was strong but still weaker than for promegestone (resulting in 62% of the promegestone efficacy, Figure 2B).

### 2.3. Cyclin D1 Expression Assay

Progesterone-induced cell proliferation might occur as cyclin D1 depending on the mechanism. Cyclin D1 in positive PR+ phenotype of T47D cells represents the target gene for progestins. PR ligated to progestin results in cyclin D1 expression linking to the intracellular signaling with mitogenic cell cycle machinery activation [15,16,17]. Therefore to estimate its overexpression inhibition in T47D cells, tested cells were treated with increasing concentration of SPRMs. Inactivation of cyclin D1 expression as well as the breast cancer cells formation repression were tested. Immunoassay with FACS analysis of cells population with endogenous cyclin D1 expression was performed. Cells treated with promegestone showed a dose-dependent induction of cyclin D1 expression.

Despite the fact asoprisnil showed a cyclin D1 expression induction tendency similar to that of promegestone, a dose-response curve revealed a weaker stimulation in the whole tested range of concentrations. No induction was observed for mifepristone with EC_50_ = 151.8 nM. Its potency relative to promegestone resulted in a factor of 1.6. A similar factor of 1.3 was obtained for ulipristal although its efficacy was equaled 78% of the promegestone activity. Efficacy parameters and EC_50_ values for mifepristone and ulipristal provide evidence that in the paracrine pathway antiprogestins act opposite to progestins in the cyclin D1 expression approach. Asoprisnil represents another trend. In spite of its low EC_50_ = 14.17 nM and strong efficacy (69.05%), Asoprisnil did not reach the full response of promegestone which may suggest a weaker than progesterone impact on cyclin D1 expression and breast cancer cell proliferation (Figure 3 and Table 2).

### 2.4. GST-Pull Down Analysis of RANK-RANKL Complex Destabilization

The presented work raises one more aspect of SPRMs’ application as a new treatment. Their new feature as inhibitors of the RANK-RANKL complex were studied. This activity was tested based on the protein-ligand interaction according to the manufacturer’s instructions of a GST-pull down assay (Thermo Scientific, Warsaw, Poland). To analyze the complex RANK-RANKL binding destruction, the competitive incubation of the RANK receptor with antiprogestin was performed for 24 h. The next day, the capture affinity of RANK by RANK-ligand previously immobilized on the agarose gel column was determined. The complex visualization was performed on SDS-PAGE gel during electrophoresis and Coomassie Blue staining. As shown in Figure 4, the complex of RANK and GST–RANKL were eluted from glutathione beads in both test and control system (lane D, part I and lane C, part II), while none of the RANK protein molecule was bound with GST- RANKL (lane C, part I) during incubation with antiprogestin. 

The results demonstrate RANK-RANKL associated blocking by antiprogestin. Additionally, the parallel performed control proved complex stability during the incubation without the presence of antiprogestin in a tested complex binding environment. The experiment carried out proved that antiprogestin destroys the RANK-RANKL complex. This class of SPRMs appears as potential anti-RANK agents with a new perspective for the problem of osteoporosis in postmenopausal women but also with potential benefits for women during breast cancer therapy that occurs with bone density loss.

## 3. Materials and Methods

### 3.1. Tested Compounds

Some SPRMs were selected so as to evaluate their activity as anti-RANK and antiproliferation agents via NFkB-cyclin D1 axis inhibition. Promegestone with agonistic activity towards progesterone receptor as well as mifepristone with antagonistic activity to progesterone receptor were used as a control. The main aim of the presented study is to estimate the potential use of new mesoprogestin (asoprisnil) and antiprogestin (ulipristal) for effective treatment of menopausal and postmenopausal symptoms like osteoporosis and breast cancer.

### 3.2. Reagents

The human breast cancer cell line T47D was obtained from Sigma-Aldrich (Poznan, Poland). The plasmids: pGL4.27 [luc2P/minP/Hygro] vector coding reporter gene of luciferase luc2P (control vector) and pGL4.32 [luc2P/NF-κB-RE/Hygro] vector coding an NF-κB response element (NF-κB-RE) were purchased from Promega (Tychy, Poland). Cell culture medium composition: DMEM; 10% FBS, 2 mM L-glutamine, 100 U/mL penicillin, 0.1 mg/mL streptomycin, 50 mg/mL higromycin were purchased from Thermo Fisher (Warsaw, Poland), Biowest (Zgierz, Poland) and Sigma-Aldrich, respectively. Ulipristal, asoprisnil, mifepristone, promegestone, and progesterone were purchased in Sigma-Aldrich. The transfection reagent Lipofectamine was obtained from Thermo Fisher Scientific. Lysate reagents M-PER Mammalian Protein Extraction Reagent and Health Protease Inhibitor Cocktail were from Thermo Fisher Scientific. The antibodies for cyclin D1 expression immunoassay analysis—mouse monoclonal IgG2-PE cyclin D1 and normal mouse IgG2a-PE antibody—came from Santa Cruz Biotechnology (Dallas, Texas, USA). In the GST-pull down analysis pure RANK and RANKL proteins were generated by Gencript (Piscataway, NJ, USA). For the WST test RANKL mouse recombinant protein and OPG human recombinant protein were purchased from Sigma-Aldrich.

### 3.3. Cell Culture

T47D cells were cultured in phenol-free DMEM medium supplemented with 10% fetal bovine serum (FBS), 2 mM L-glutamine, and 100 U/mL penicillin/streptomycin. Cultures were fed every 2–3 days by replacing with fresh medium.

### 3.4. The proliferation Assay

T47D cells were seeded in 96-well plates at a density of 8 × 10^3^ cells per well with three replicates and stimulated with RANKL (50 ng·mL^−1^). Three days later, cells were treated with the increasing concentrations of SPRMs (1.17–1200 nM) and 100 ng·mL^−1^ of OPG for 24 h post-culture. The medium was refreshed into one with charcoal FBS. On the 5th day, WST reagent was added and the absorbance was measured. Based on the previously determined calibration curve, the number of cells per well was calculated and expressed in concentration dependent manner.

### 3.5. Estimation of NFkB Expression

#### 3.5.1. Cell Test System

The initial cells density (0.5 × 10^3^ cells per well) were grown on 6-well plates in transfection medium w/o antibiotics. Only confluent cells were used for transfection with NFkB DNA using Lipofectamine according to the manufacturer’s instruction. Next, the stable transfectants were cultured in the growth medium containing antibiotic (hygromycin 600 µg/mL). The selection of stably transfected cells was carried out based on the resistance to hygromycin, which came to the transfected cells together with the NFkB DNA. It meant, that only cells growing in such a medium were able to carry a gene of selection [18].

#### 3.5.2. Transactivation Assay

Briefly, cells were seeded into 96-well plates and maintained in cell culture media (phenol-free DMEM and charcoal FBS) for 24 h. Cells were then pre-treated with the increasing concentrations of SPRMs (1–1200nM) (ulipristal U and asoprisnil—A) and reference compounds mifepristone (M), and promegestone (P) for 24 h, with or without RANKL stimulation (50 ng·mL^−1^), respectively. Luciferase activity was measured using the Promega Luciferase Assay System and normalized to that of the vehicle control.

### 3.6. Flow Cytometry Analysis of Cyclin D1 Expression

#### Cell Test System

T47D cells (0.5 × 10^6^ cells/well) were cultured in 6-well plates in a medium. Twenty-four h later, the attached cells were treated with tested compounds. Compounds were used at six different concentrations: 1.17, 19, 75, 300, 600 and 1200 nM. After 24 h cells were detached and transferred into centrifuge tubes (1–2 × 10^6^ sample) and spun at 400× *g* for 8 min at room temperature. Next cells were fixed in 22.5 µL of PBS with 157.5 µL of 80% ethanol (cold, chilled at −20 ℃, final conc. of ethanol was 70%) and incubated overnight at 4 ℃. After washing them in PBS, 150 µL of 3% BSA/PBS was added to the sample and cells were incubated for 30 min at 4 ℃. Next immunolabeling was performed. Incubation of 100 µL of previously prepared antibody recognizing cyclin D1 (anti cyclin D1 Ab – PE conjugated, 1:50 dilution in 3% BSA/PBS) with 1 × 10^6^ cells/sample lasted 30 min at 4 ℃ and was protected from direct light. For negative control, 100 µL of isotype control IgG – PE conjugated (diluted the same way as first antibody) was used. Gently resuspended cells were incubated for 30 min at 4 ℃, sample was protected from light as well. Next cells were washed two times in 300 µL of PBS and finally, cells were resuspended in 500 µL of DPBS and flow cytometry measurement was performed. A CytoFLEX flow cytometer, (Beckmann Coulter, Warsaw, Poland) was used for analysis with the following settings: 10,000–30,000 gated cells were analyzed for each sample. The expression of the cyclin D1 for each sample was analyzed by the Kaluza Analysis Software, ver. 1.5a (Beckmann Coulter).

### 3.7. GST-Pull Down Assay for RANK/RANKL Complex Binding

The binding interaction of RANK and RANKL was tested in a GST-pull down assay. Firstly 100 nM 0.1% DMSO stock solution of antiprogestin was prepared. Prepared samples of RANK and RANKL contained 100 µg of each protein. The incubation of RANK and antiprogestin mixture lasted 24 h. First, RANKL fused with GST was immobilized on glutathione agarose resin at 4 °C for 60 min in a Pierce spin column. Next, 800 µL of prepared RANK solution potentially bonded with tested antiprogestin was added to the column containing the immobilized RANKL and incubated for the next 120 min with gentle rocking motion on rotating platform to let the complex building or its destruction occur. After washing of unassociated RANK, the bounded RANK and RANKL were eluted by addition elution buffer. The eluted proteins as well as the complex was run on 12% SDS–PAGE and stained with Coomassie Blue.

## 4. Conclusions

The RANK/RANKL factors are still being intensively studied. Nowadays their function in bone metabolism and breast cancer proliferation is well documented. The mechanism of RANK/RANKL stimulation is known and widely described in many papers. Progesterone is the ligand of the progesterone receptor (PR) and it is known to trigger RANK expression, which enhances the proliferation of different cells such as osteoclasts and epithelial breast cells. In this way, the therapy based on the administration of the agonistic ligands of the sex hormone receptors leads to different strong and serious sides effects [18,19,20,21,22,23,24]. A new class of compounds which would be used as anti-RANK agents are selective progesterone modulators – mesoprogestins and antiprogestins. Asoprisnil was tested in both the endocrine and paracrine pathways with NFkB and cyclin D1 expression- stimulated cells division with the same force (62% and 68% of promegestone efficacy, respectively). This data confirms its weaker interaction either with progesterone and RANK receptor. Ulipristal at EC_70_ = 21 nM limited NFkB expression via an endocrine pathway with effectiveness equal to 27% of the efficacy of promegestone and subsequently reduced cyclin D 1 expression in a paracrine pathway. Based on the obtained results, the main achievement of the presented work is to propose the new generation of antiprogestins as promising candidates for the effective treatment of over proliferative tissues like breast cancer and osteoclasts in osteoporosis. Additionally, based on the performed positive first attempt to prove a direct interaction of antiprogestin with RANK with regards to RANKL competitive binding to RANK, antiprogestins arise as interesting compounds that may play an important role in the treatment of osteoporosis in hormonal-dependent diseases. Furthermore, their unique mechanism of downstream RANK-RANKL signaling might guarantee the prevention of breast cancer and diminish the metastasis of breast cancer to bone.

## Figures and Tables

**Figure 1 molecules-25-01321-f001:**
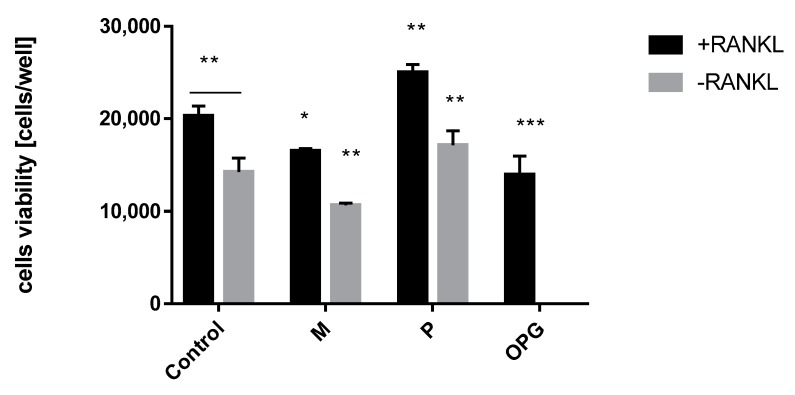
+RANKL and -RANKL induced T47D cells proliferation upon mifepristone (M, 1200 nM), progesterone (P, 1200 nM) and OPG treatment. Data represent the mean ± SD (*n = 3*). *p* values correspond to controls with statistically significant differences: * *p* ≤ 0.05, ** *p* ≤ 0.01, *** *p* ≤ 0.001 (two-way ANOVA variance, multiple comparison of the tested groups vs. control group, separately for +RANKL and -RANKL conditions).

**Figure 2 molecules-25-01321-f002:**
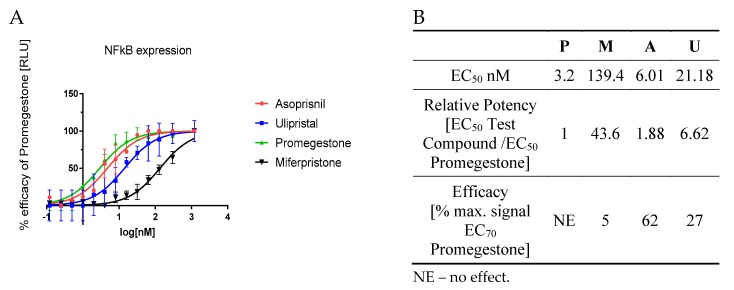
NFkB expression analysis upon SPRM activity in transcription and transactivation manner (**A**) graph presents mean ± SD, *n* = 3; (**B**) EC_50_ and EC_70_ and efficacy parameters), P-Promegestone, M-Mifepristone, A-Asoprisnil, U-Ulipristal.

**Figure 3 molecules-25-01321-f003:**
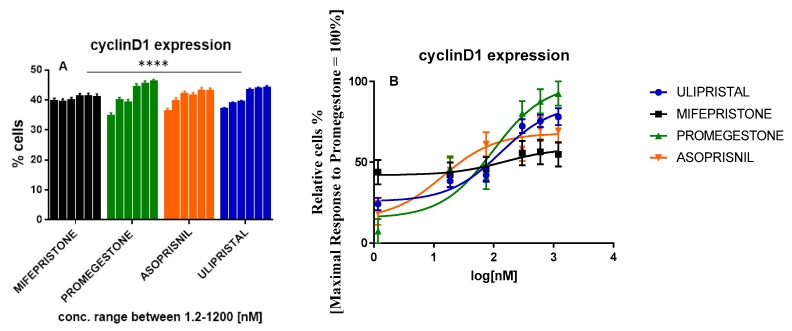
The tested compounds induced cyclin D1 expression in T47D cells during treatment for 24 h; (**A**): FACS analysis of cells population with cyclin D1 expression, data represent three separate experiments, *n = 3*, mean ± SD with statistic significant ***** p* ≤ 0.0001 (two-way ANOVA for comparison of tested groups against each other, mean volumes for each concentration were analyzed, data for the control group although statistically significant, there was not published); (**B**): normalized % of cells with cyclin D1 expression to maximal % of cells expressing cyclin D1 upon Promegestone stimulation.

**Figure 4 molecules-25-01321-f004:**
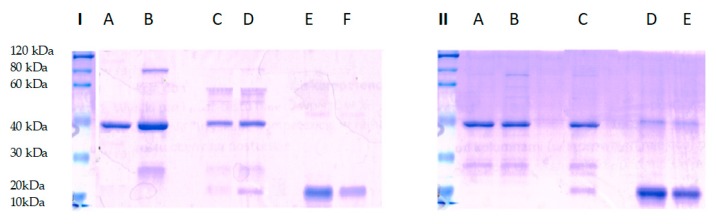
SDE-PAGE analysis of RANK-RANKL complex stabilization. I test system: Line A—RANKL protein control, Line B—RANKL after column washing; Line C—complex destruction after 24 h incubation of RANK with antiprogestin, Line D—complex binding after 24 h incubation of RANKL and RANK; Line E—RANK protein after column washing; Line F—RANK protein control. II control system: Line A—RANKL protein control, Line B—RANKL after column washing; Line C—complex binding after 24 h incubation of RANKL with RANK, Line D—RANK protein after washing column; Line E—RANK protein control. The protein ladder comes from separate run.

**Table 1 molecules-25-01321-t001:** Structure of tested progestins and SPRMs.

Progestins		SPRMs		
Progesterone	Promegestone	Mifepristone	Ulipristal	Asoprisnil
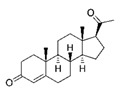	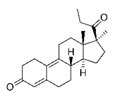	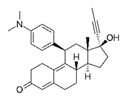	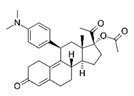	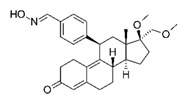

**Table 2 molecules-25-01321-t002:** Cyclin D1 expression induced by SPRMs in T47D cells (EC_50_, relative potency and efficacy parameters).

	Promegestone	Mifepristone	Asoprisnil	Ulipristal
EC_50_ nM	56.22	113	20.25	94.33
Relative Potency [EC_50_ Test Compound/EC_50_ Promegestone]	1	2.0	0.36	1.67
Efficacy [% max. signal EC_50_ Promegestone]	NE	23	82	27

NE-no effect.

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
