# Peer review of "New Selective Progesterone Receptor Modulators and Their Impact on the RANK/RANKL Complex Activity"

_molecules, 2020, doi:10.3390/molecules25061321_

Round 1

Reviewer 1 Report

The research presented in this paper is interesting and generally the experiments are well designed. However the English is so weak that sometimes is difficult to understand the work( methods, discussion and interpretation of the results). Frequently we need to read it twice to understand. I suggest a complete reformulation of the overall writing in order to clear and put it more understandable.

Line 9: The first phrase of the abstract is inadequate  for the subject of the paper

line 36-39: rewrite the phrase...not clear

line 42: Estradiol deficiently?? ..reformulate

line 43: correct the verb

line 48: ...indirectly...

line 57: described in...

line 58: find another verb to replace "liganded"

Please include one figure with the chemical structures of tested compounds ulipristal and asoprisnil as well as mifepristone and promesgestone

line 81...compared...

line 90...antiprolifer...misspelling

line 94: cell viability is expressed in what...????

line 115 invert the order of the figures the first referred must be 2A not 2B...

line 117 antiprogestins...misspelling

line 120 Why EC70...and not EC90 or EC50...explain

line 143 in figure 3A add the concentrations in the X line of the bars

line 189: in II test system the F line- rank protein control is not visible...

line 203 and 204 ...misspelling glutamine, penicillin, etc

line 222: in the description of the  cell proliferation assay you say that thw assement was done by the reading of absorvance but with 10000, 20000...strange...

line 295: some bibliographic references are not in English... ref 1...3.

Author Response

Referring to the Reviewer 1 comment, the author response has been highlighted red:

Line 9: The first phrase of the abstract is inadequate  for the subject of the paper

It has been adjusted to the subject of the paper.

line 36-39: rewrite the phrase...not clear

It has been corrected and clarified.

line 42: Estradiol deficiently?? ..reformulate

The word ‘‘deficiently’’ has been changed into ‘’deficiency’’.

line 43: correct the verb

The verb has been corrected.

line 48: ...indirectly...

It has been corrected.

line 57: described in...

It has been corrected.

line 58: find another verb to replace "liganded"

‘’Liganded’’ has been replaced by ‘’bound up with’’.

Please include one figure with the chemical structures of tested compounds ulipristal and asoprisnil as well as mifepristone and promegestone

The table 1 with structures of the tested compounds has been added to the paper.

line 81...compared...

The word has been corrected and used in proper grammar form.

line 90...antiprolifer...misspelling

Misspelling has been corrected.

line 94: cell viability is expressed in what...????

Cells viability has been expressed as cells per well.

line 115 invert the order of the figures the first referred must be 2A not 2B...

The description and label of the figures have been corrected.

line 117 antiprogestins...misspelling

Misspelling has been corrected.

line 120 Why EC70...and not EC90 or EC50...explain

The misspellings have been corrected.

line 143 in figure 3A add the concentrations in the X line of the bars

The concentration points have been described in section materials and methods, section 3.6-cell test system.

line 189: in II test system the F line- rank protein control is not visible...

The description and label of lines have been clarified.

line 203 and 204 ...misspelling glutamine, penicillin, etc

Misspellings have been corrected.

line 222: in the description of the  cell proliferation assay you say that thw assement was done by the reading of absorvance but with 10000, 20000...strange...

Based on a previously determined calibration curve, the number of cells per well was calculated and expressed in concentration dependent manner. Y-axis on figure 1 has been adjusted to the performed analysis.

line 295: some bibliographic references are not in English... ref 1...3.

The titles of references 1-3 have been written down in English.

Reviewer 2 Report

The author should rewrite the manuscript. There are numerous mistakes and inconsistent writing. Define the acronyms at the beginning of the introduction.

Author Response

Referring to the Reviewer 2 comment, the author response has been highlighted red:

The author should rewrite the manuscript. There are numerous mistakes and inconsistent writing. Define the acronyms at the beginning of the introduction.

The grammar mistakes have been corrected.

The text has been improved.

The acronyms have been defined in the text.

Reviewer 3 Report

The manuscript entitled” New selective progesterone modulators and their impact on the RANK/RANKL complex activity” reported induction of cell proliferation via RANK/RANKL signaling pathway by new selective progestin modulators. In this manuscript, the authors tried to measure the antiproliferation effect of SPRMs with WST assay, potential mechanism of SPRMs on proliferation and direct effect of SPRMs on RNAK/RANKL complex with GST-pull down analysis. The results are very interesting; however, this manuscript may not be suitable to Molecules in present form.

Below are the detailed comments for authors.

1, the ambiguous scale bar used for Y axis in figure1 is not appropriate.

2, NF-kappaB may consist of different subunits, such as p100/RelB or p65/p50, the authors should indicate which subunits of NF-kappaB DNA was overexpressed and why only this form was measured.

3, it seems that the author would like to propose that SPRMs may inhibit cell proliferation via NFkB-cyclin D1 axis, although it is highly possible, expression data of both NF-kappaB and cyclin D in the cells (suppression of NF-kappaB then check expression of cyclin D after stimulation with SPRMs, etc.) would provide better support to this hypothesis.

4, some minor revision: line 90: “Positive aspect of aniproliferation activity….”, I guess the “aniproliferation” should be antiproliferation. Line 246, “Next cells were lysated in 22.5 µL of PBS with 157.5….” I suppose the “lysated” may refer to “fixed”.

Author Response

Referring to the Reviewer 3 comment, the author response has been highlighted red:

1, the ambiguous scale bar used for Y axis in figure1 is not appropriate.

The Y axis in figure 1 has been clarified.

2, NF-kappaB may consist of different subunits, such as p100/RelB or p65/p50, the authors should indicate which subunits of NF-kappaB DNA was overexpressed and why only this form was measured.

Details referring to the construct of NFkB vector have been indicated in the section materials and methods, point 3.2 Reagents.

The pGL4.32[ luc2P /NF-κB-RE/Hygro] vector contains five copies of an NF-κB response element (NF-κB-RE) that drives transcription of the luciferase reporter gene luc2P ( Photinus pyralis ). Luc2P is a synthetically derived luciferase sequence with humanized codon optimization that is designed for high expression and reduced anomalous transcription.

3, it seems that the author would like to propose that SPRMs may inhibit cell proliferation via NFkB-cyclin D1 axis, although it is highly possible, expression data of both NF-kappaB and cyclin D in the cells (suppression of NF-kappaB then check expression of cyclin D after stimulation with SPRMs, etc.) would provide better support to this hypothesis.

NFkB expression was tested in a form of clonal stable reporter cell line established on the unified cellular background in T47D cells. This cell line was prepared by introducing the expression vector encoding NFkB corresponding to pGL4.32 with luciferase reporter vector. Cells that stably integrated construct into the genome were isolated with selection medium containing hygromycin and validated with the collection of positively transfected clones.

Cyclin D was tested in a form of endogenic background in T47D cells. The colony of T47D cells was treated by tested compounds and cyclin D expression was determined during immuno-assay.

Since separate signaling pathways have been planned and studied, there it does not make sense to analyze them both at the same time since there was not any common target point which would have prescribed information from one to another pathway.

4, some minor revision: line 90: “Positive aspect of aniproliferation activity….”, I guess the “aniproliferation” should be antiproliferation. Line 246, “Next cells were lysated in 22.5 µL of PBS with 157.5….” I suppose the “lysated” may refer to “fixed”.

The misspellings have been corrected.

Round 2

Reviewer 2 Report

The authors have amended the manuscript but still most of the sentences are too long to understand. Some of the concerns in the introduction are ...

  1. Define RANK/RANKL in line '34'
  2. How can anyone claim the most popular disease without a citation? Change the sentence, lines '40-42'.
  3.  The sentence in lines '58-60' is not clear. 'When receptor RANK bound up with RANKL, binds to molecule TRAF6 (TNF 59 receptor-associated factor 6), activates the kinases complex mainly through activation of 60 the mitogen-activated protein kinase (MAPK) and NF-κB by IκB kinase (IKK).'

Author Response

Referring to the Reviewer 2 comment – round 2, the author response has been highlighted red:

1. Define RANK/RANKL in line '34'

RANK and RANKL proteins have been defined as follows:

RANK - Receptor activator of nuclear factor κ B , there is receptor for RANK-Ligand.

RANKL- Receptor activator of nuclear factor kappa-Β ligand.

2. How can anyone claim the most popular disease without a citation? Change the sentence, lines '40-42'.

The suitable citation has been included and the sentence has been changed.

3.  The sentence in lines '58-60' is not clear. 'When receptor RANK bound up with RANKL, binds to molecule TRAF6 (TNF 59 receptor-associated factor 6), activates the kinases complex mainly through activation of 60 the mitogen-activated protein kinase (MAPK) and NF-κB by IκB kinase (IKK).'

Since the mentioned fragment does not suit the text context, therefore so as to improve text readability, the author has decided to delete this fragment.

Reviewer 3 Report

2. The question is not ask which vector used, but which subunits were measured.

3. The question is not to ask whether NFkB and Cyclin D should be measured together, but to ask NFkB should be measured to make sure the transfection was effective.

Author Response

Referring to the Reviewer 3 comment, round 2, the author response has been highlighted red:

  1. The question is not ask which vector used, but which subunits were measured.

Since the aim of the study was to analyze NFkB expression using vector with NFkB RE, therefore analysis of the appropriate subunit p65, RelB or c-Rel overexpression was not performed. Basically, p65 or complex of RelA-p50 are most abundant transcriptional factors.

  1. The question is not to ask whether NFkB and Cyclin D should be measured together, but to ask NFkB should be measured to make sure the transfection was effective.

As far as the transfection efficiency, the stable transfectants were selected based on the hygromycin resistance. Additionally, pGL4.27 [luc2P / minP / Hygro] vector was used as a negative control.